# Methyl-Jasmonate Functions as a Molecular Switch Promoting Cross-Talk between Pathways for the Biosynthesis of Isoprenoid Backbones Used to Modify Proteins in Plants

**DOI:** 10.3390/plants13081110

**Published:** 2024-04-16

**Authors:** Quentin Chevalier, Alexandre Huchelmann, Pauline Debié, Pierre Mercier, Michael Hartmann, Catherine Vonthron-Sénécheau, Thomas J. Bach, Hubert Schaller, Andréa Hemmerlin

**Affiliations:** 1Centre National de la Recherche Scientifique, Institut de Biologie Moléculaire des Plantes (IBMP), Université de Strasbourg, 12 rue du Général Zimmer, F-67084 Strasbourg, France; qchevalier67@gmail.com (Q.C.); pauline.debie2@etu.unistra.fr (P.D.); pierre.mercier@ibmp-cnrs.unistra.fr (P.M.); mhartmann@plant-biotech.com (M.H.); bach@unistra.fr (T.J.B.); hubert.schaller@ibmp-cnrs.unistra.fr (H.S.); 2Centre National de la Recherche Scientifique, Laboratoire d’Innovation Thérapeutique, Université de Strasbourg, CEDEX, F-67401 Illkirch, France; vonthron@unistra.fr

**Keywords:** metabolic cross-talk, isoprenoid biosynthesis, isoprenylated proteins, jasmonic acid methyl esther/MeJA, MEP pathway, MVA pathway

## Abstract

In plants, the plastidial mevalonate (MVA)-independent pathway is required for the modification with geranylgeranyl groups of CaaL-motif proteins, which are substrates of protein geranylgeranyltransferase type-I (PGGT-I). As a consequence, fosmidomycin, a specific inhibitor of 1-deoxy-d-xylulose (DX)-5 phosphate reductoisomerase/DXR, the second enzyme in this so-called methylerythritol phosphate (MEP) pathway, also acts as an effective inhibitor of protein prenylation. This can be visualized in plant cells by confocal microscopy by expressing GFP-CaM-CVIL, a prenylation sensor protein. After treatment with fosmidomycin, the plasma membrane localization of this GFP-based sensor is altered, and a nuclear distribution of fluorescence is observed instead. In tobacco cells, a visual screen of conditions allowing membrane localization in the presence of fosmidomycin identified jasmonic acid methyl esther (MeJA) as a chemical capable of gradually overcoming inhibition. Using Arabidopsis protein prenyltransferase loss-of-function mutant lines expressing GFP-CaM-CVIL proteins, we demonstrated that in the presence of MeJA, protein farnesyltransferase (PFT) can modify the GFP-CaM-CVIL sensor, a substrate the enzyme does not recognize under standard conditions. Similar to MeJA, farnesol and MVA also alter the protein substrate specificity of PFT, whereas DX and geranylgeraniol have limited or no effect. Our data suggest that MeJA adjusts the protein substrate specificity of PFT by promoting a metabolic cross-talk directing the origin of the prenyl group used to modify the protein. MVA, or an MVA-derived metabolite, appears to be a key metabolic intermediate for this change in substrate specificity.

## 1. Introduction

Type I protein prenylation is a post-translational process common to eukaryotes that adds a hydrophobic farnesyl (F) or geranylgeranyl (GG) moiety to proteins harboring a C-terminus CaaX (C = cyst; a = aliphatic; X = C-terminus amino acid) motif. Modified proteins attain new traits, mainly hydrophobicity. Thus, prenylation enables interactions with membrane structures, with other proteins, for example, in mega-complexes, or changes the conformation of the target protein. It relies on two metalloenzymes that have to discern two co-substrates: the CaaX protein and a prenyl diphosphate. These protein prenyltransferases (PPTs) are heterodimers with a common α-subunit and a distinct and specific β-subunit. Accordingly, protein farnesyltransferase (PFT; EC 2.5.1.58) catalyzes a thioetherification between the cysteine and farnesyl using farnesyl diphosphate (FPP) as a co-substrate, while type-I protein geranylgeranyltransferase (PGGT-I; EC 2.5.1.59) uses geranylgeranyl diphosphate (GGPP). It had been accepted that the nature of the “X” amino acid guides the enzyme’s reaction to either a transfer of farnesyl (farnesylation) catalyzed by PFT or geranylgeranyl (geranylgeranylation) catalyzed by PGGT-I [1]. Although, in practice, very few prenylated proteins have been characterized in plants, it is clear that CaaX-motif proteins are involved in many cellular processes. They are particularly involved in plant development and stress responses [2,3]. The use and characterization of loss-of-function mutants have been fundamental to these investigations. In this regard, Arabidopsis T-DNA insertion mutants in the *PLP* gene encoding the α-subunit, in the *ERA1* gene (*Enhanced Response to ABA-1*) encoding the β-subunit of PFT, and in the *GGB* gene encoding the β-subunit of PGGT have been described [4,5,6,7]. The *ggb* mutant plant lacks any evident developmental phenotype but shows an enhanced response to abscisic acid (ABA) in stomatal closure assays [7]. Compared to *ggb* plants, the phenotypes of Arabidopsis *era1* mutant plants are more obvious, with an enhanced response to ABA in stomatal closure of guard cells but also in seed germination. Furthermore, *era1* mutants are characterized by stunted growth and reduced fertility under standard growth conditions. In addition, their meristems are enlarged, and they exhibit supernumerary floral organs as well as larger seeds [8,9,10,11,12]. Moreover, compared to wild-type plants, *era1* loss-of-function plants display different sensibilities towards environmental factors with clear drought and heat tolerance phenotypes [4,5,13,14,15,16,17]. The phenotypes of the *plp* mutant [6] and the *era1ggb* double mutant [7] are comparable to those of *era1*, but significantly more marked. Both mutant plants produce severely hypertrophied meristems and additional floral organs and show severe growth defects with a significantly reduced stature compared to the wild-type plant [6,7].

To achieve biological activity, CaaX-proteins need to be modified with the prenyl group, which ensures proper subcellular localization. Moreover, it is noteworthy that the proteolysis process that results in the release of the ‘aaX’ peptide and the carboxymethylation of the cysteine are crucial steps for biological activity, at least for some proteins [18]. The modulation of PPT activity can be achieved at two distinct levels: either by an extension of protein substrates used specifically by an enzyme or by using alternative prenyl diphosphate substrates to modify the protein. The CaaX consensus motif (containing three variable residues) is by definition not very stringent and has over the years constantly extended and reevaluated [19]. A further degree of flexibility, especially in plants, is achieved by the use of the second substrate, prenyl diphosphate. Since FPP and GGPP are central isoprenoid precursors shared by different metabolic branches in the pathway, PPTs must compete with other enzymes for the availability of these substrates. For instance, in *Nicotiana tabacum*, phytoalexins, including capsidiol, lubimin, rishitin, solavetivone or phytuberin [20], are sesquiterpenoids derived from FPP. In a similar way, GGPP is, for instance, the precursor of plastidial pigments such as carotenoids and chlorophylls. It is noteworthy to mention that the concentrations of prenyl diphosphates needed for protein modifications are much lower than those needed for sesquiterpenoid or pigment biosynthesis. The way PPTs obtain their part of the cake must be closely connected to metabolic regulation, cellular compartmentation, and the affinity of enzymes to their substrate. However, the molecular mechanisms by which such a regulation occurs in vivo are very poorly understood, with most of the data being generated from experiments performed in vitro [21]. Protein prenylation in mammals and yeast relies exclusively on the classical mevalonic acid (MVA) isoprenoid biosynthesis pathway [1]. In contrast, plants are unique in that they use two isoprenoid biosynthesis pathways in parallel, with the potential for exchanging metabolic intermediates. This flexibility is known as metabolic cross-talk for the biosynthesis of isoprenoids (see [22]). In essence, modifications with these hydrophobic prenyl groups depend on isoprenoid precursors (FPP or GGPP) biosynthesized either through the classical MVA pathway [23,24,25,26] or the plastidial methylerythritol phosphate (MEP) pathway [27,28,29]. To study the regulation of protein prenylation in vivo, this characteristic is of major advantage in determining the metabolic origins of prenyl groups used to modify prenylated proteins [29]. Using specific inhibitors for the synthesis of GGPP (from the MEP pathway) or FPP (from the MVA pathway) enables discrimination of the metabolic origin under specific conditions. Furthermore, instead of being neosynthesized, FPP can be recycled through a process known as the “FPP salvage pathway,” implemented for modifying a new series of proteins (for review, see [30]). Plants also generate a broader range of prenyl diphosphates in comparison to other organisms using type I protein prenylation [31]. Chain length and saturation degree of those prenyl diphosphates fluctuate [32]. Interestingly, some of those (e.g., dolichols, phytyl residues) have been proposed to be incorporated into prenylated proteins [25]. Jointly, these singularities in plants allow precursor supply to be adjusted to specific biosynthetic needs and provide the flexibility to biosynthesize a variety of different isoprenoid compounds simultaneously. Although extensive research has been conducted to study metabolic cross-talk for the biosynthesis of isoprenoid metabolites [22,33,34,35], the situation regarding the supply of FPP and GGPP used in protein modification is barely described. Tobacco BY-2 cells have been widely used as a model plant for the study of protein prenylation in plants for several reasons. They divide rapidly and thus are metabolically very active, but also because the catalytic activities of PPTs are high [23]. Under standard culture conditions, this cell suspension uses exclusively plastidial MEP-derived GGPP to modify a chimeric GFP-CaM-CVIL protein that can be defined as a prenylable GFP-based sensor [28]. Prenylation can be visualized by microscopy and is characterized by a reallocation of the fluorescent signal from the plasma membrane into the nucleus when cells are treated with MEP pathway-specific inhibitors, like fosmidomycin or oxoclomazone [28,36]. Thus, the PPT that catalyzes the transfer of GGPP to the cysteinyl residue exclusively uses a C_20_ substrate derived from MEP and is unable to substitute this substrate with a prenyl group derived from MVA. The question arising is whether, under specific conditions, prenylation of this protein can be achieved using MVA-derived isoprenyl units. A switch at this level would indicate the existence of specific signals that initiate metabolic cross-talk in the cell. Yet, in this context, it has been shown that farnesol (Fol) stimulates plant 3-hydroxy-3-methyl glutaryl coenzyme A reductase (HMGR) activity, the key enzyme of the MVA pathway [37]. Interestingly, this prenol functions as a compound shifting the metabolic origin of the prenylation substrates from the MEP to the MVA pathway, which modifies the GFP-CaM-CVIL sensor protein [38]. The mechanism by which this change occurs is unknown and needs further investigation. The identification of cross-talk inducers within cellular systems would aid in the establishment of optimal conditions for the accumulation of valuable isoprenoid metabolites in plants. Thus, we screened for chemicals that were able to allow protein prenylation in tobacco BY-2 cells under restrictive GGPP biosynthesis conditions. The effect of the stress hormone jasmonic acid methyl esther (methyl-jasmonate, MeJA), which was identified as a key element in the induction of such a metabolic switch, was carefully investigated and analyzed to figure out whether the prenylation activity switches from the use of a MEP pathway-derived metabolite to that of an MVA-derived one.

## 2. Results

### 2.1. MeJA Promotes Membrane Localization of Prenylated MEP-Derived CaaL-Box Proteins in the Presence of Fosmidomycin

Our prior research shows that, under standard growth conditions, the MEP pathway supplies prenyl diphosphate used to modify GFP-CaaL-like proteins in tobacco cells and plants [28,29,36,38,39]. With this in mind, the objective of this study was to identify conditions under which the MVA pathway could be activated to take over the supply of prenyl diphosphates in order to modify a protein substrate typically dedicated to PGGT.

To do so, the prenylation sensor protein GFP-CaM-CVIL expressed in the tobacco BY-2 cell line under the control of an inducible dexamethasone promoter has been used [28]. Under standard growth conditions, GFP-CaM-CVIL is geranylgeranylated and localizes to the plasma membrane (Figure 1a, CONTROL). When treated with the MEP pathway-specific inhibitor fosmidomycin, which blocks plastidial GGPP biosynthesis by inhibiting the second enzyme in the MEP pathway (1-deoxy-d-xylulose 5-phosphate reductoisomerase), the fluorescent protein labels the nucleus, triggered by a nuclear localization sequence (NLS) motif (Figure 1a, FOS). This indicates a deficiency in prenylation [28]. We ran an optical screen for compounds preventing nuclear localization of the fluorescent GFP-BD-CVIL protein in the presence of fosmidomycin. The candidate compound MeJA (50 μM) restored plasma membrane labeling (Figure 1a, FOS + MeJA). Since cells were co-treated with rather high concentrations of fosmidomycin (100 μM), it can be inferred that MEP pathway-independent prenylation has been activated. To assess the reproducibility of the effect of MeJA, statistical analyses were conducted, and we also used a tobacco BY-2 cell line expressing GFP-ROP6-CSIL, another prenylated protein. ROP6-CSIL is not only prenylated but is also acylated [40]. MeJA (20 μM), as the only chemical added to the culture, had no significant effect on GFP-CaaL prenylation. Fosmidomycin (100 μM) delocalizes proteins into the nucleus (>87%), and in combination with MeJA, full membrane localization (93% and 65%) was significantly reached (Figure 1b). In order to evaluate the minimum concentration of MeJA required to achieve protein prenylation, a concentration range from 0.5 μM up to 20 μM was examined (Figure 1c). Concentrations as low as 0.5 μM MeJA lead to significant relocalization of GFP-CaM-CVIL into the plasma membrane (Figure 1c). In a dose-dependent manner, the higher the concentration of MeJA, the better the membrane localization is detected. Other phytohormones have been tested, but in the presence of ABA, IAA, NPA, or kinetin, inhibition with fosmidomycin remained efficient (Appendix A). However, ethephon (2-chloroethylphosphonic acid), which is an ethylene-releasing agent, has also been identified (Appendix A). As both MeJA and ethylene are defense signaling hormones [41], we wondered whether ethylene could act as a potentiator with MeJA. At a concentration of 1 μM MeJA, a cumulative effect overcoming inhibition by 100 μM fosmidomycin was observed (Figure 1d), supporting the idea that ethylene acts synergistically with MeJA.

### 2.2. PFT Modifies GFP-CaM-CVIL in the Presence of MeJA

MeJA-induced relocalization of GFP-CaM-CVIL in the presence of fosmidomycin raises new questions, specifically: what enzyme is responsible for catalyzing the transfer of a prenyl group in treated cells? To answer this question, we analyzed the subcellular localization of GFP-CaM-CVIL in Arabidopsis KO mutant plants, in which only one PPT type I is functional: either PGGT-I in *era1* lines or PFT in *ggb* lines. The dexamethasone-inducible expression of GFP-CaM-CVIL is compared to that of a transformed Col-0 line (Control line). The dexamethasone-induced subcellular localization of GFP-CaM-CVIL was monitored by confocal microscopy in leaves (surrounded by a green rectangle) and roots (surrounded by a yellow rectangle) of two-to-three-week-old seedlings (Figure 2, Appendix A). As expected, the membrane localization observed in root and epidermal cells of Col-0 (wild-type/WT) or *era1* lines is consistent with PGGT-I being the active enzyme catalyzing the prenylation of GFP-CaM-CVIL under standard growth conditions (Figure 2; Appendix A). Furthermore, a nuclear localization of GFP-CaM-CVIL expressed in *ggb* mutant lines indicates that in the absence of an active PGGT, PFT is unable to take over prenylation of the sensor protein (Figure 2; Appendix A). This mutant line, which is the only one with a nuclear localization of GFP-CaM-CVIL, has been used to test the effect of MeJA. In contrast to untreated *ggb* plants, in the presence of MeJA (30 μM), the protein localizes mainly in the plasma membrane, with only negligible nuclear labeling (surrounded by a blue rectangle, Figure 2). From this, it can be deduced that in the presence of MeJA, PFT, which is the active enzyme in *ggb* plants, recognizes the GFP-CaM-CVIL substrate typically dedicated to PGGT-I, and that MeJA seems to induce a signal directing PFT substrate specificity. The question that arises is: what kind of signal might be responsible for this functional enzymatic adjustment?

### 2.3. MeJA Enhances Protein Prenylation Capacity in Tobacco

MeJA induces both the production of MVA- and MEP pathway-derived metabolites [22,42,43]. With this in mind, we hypothesized that MeJA may affect the metabolic flux leading to the biosynthesis of prenyl diphosphates used to modify proteins. To test this hypothesis, we analyzed the subcellular localization of GFP-CaM-CVIL expressed in tobacco (*Nicotiana tabacum* L. var. xanthi) plants. Tobacco plants have been chosen because elicitation induces isoprenoid metabolite production. We established experimental conditions by using cellulase-induced elicitation (Appendix A), which enhances MVA biosynthesis via HMGR induction, ultimately leading to the biosynthesis of capsidiol, a sesquiterpenoid phytoalexin [39]. Previously, we routinely used 0.5% cellulase to set up elicitation experiences [39], and here we tested whether the treatment could modify the efficiency of prenylation.

Like tobacco BY-2 cells or Arabidopsis leaves and roots, untreated tobacco epidermal cells show plasma membrane labelling (Figure 3a,d). In this experiment, we opted to block both the MVA and MEP pathways simultaneously in order to prevent the biosynthesis of isoprenoid precursors. Thus, we determined minimal concentrations of fosmidomycin (FOS/blocking the MEP pathway) and mevinolin (MV/blocking the MVA pathway), leading to the delocalization of GFP-CaM-CVIL into the nucleus (Appendix A). By evaluating the nuclear localization in FOS and MV-combined treatments, we selected mevinolin (5 μM) and fosmidomycin (100 μM) as reference concentrations (FOS100/MV5) that were sufficient to block protein prenylation in leaf disks. It has to be noted here that without inhibition of the MVA pathway, at 300 μM, fosmidomycin delocalizes the protein into the nucleus, while mevinolin, even at very high concentrations, cannot. To test whether stress promotes prenylation efficiency in tobacco leaf disks, cellulase and MeJA treatments were used to assess membrane localization in the presence of FOS100/MV5 (Figure 3b,e). Membrane localization is restored when inhibition of prenylation is combined with cellulase treatments (Figure 3c). and with MeJA treatments (Figure 3f). 

These results suggest that the restoration of protein prenylation is due to an excess of prenyl diphosphate substrate generated in response to cellulase-induced elicitation. Cellulase-induced elicitation is associated with the stimulation of HMGR activity, which is necessary for the production of MVA used in the synthesis of capsidiol [39]. We therefore hypothesized that MVA production could act as a signal to reverse inhibition by fosmidomycin in tobacco BY-2 cells (Figure 1).

### 2.4. GFP-CaM-CVIL Prenylation under Restrictive Conditions Is Correlated with Mobilization of the MVA Pathway

To support our assertions, we subsequently examined the impact of another compound that is known to stimulate HMGR activity by acting as an elicitor. As mentioned earlier, Fol has been described as a chemical stimulating HMGR in tobacco BY-2 cells, both at a transcriptional and translational level [37]. This results in a rise in MVA levels, the enzyme product. At the same time, Fol is a substrate of specific kinases catalyzing the formation of FPP in the realm of the FPP salvage pathway [30]. In addition, Fol can overcome inhibition by fosmidomycin in tobacco BY-2 cells, but not when cells are inhibited with fosmidomycin and mevinolin simultaneously [38]. It was concluded that the MVA pathway is involved in protein modification under those conditions. On this basis, we tested the ability of Fol to induce membrane localization of GFP-CaM-CVIL expressed in Arabidopsis *ggb* lines, where only PFT is active (Figure 4). At 20 μM, Fol partially restores membrane localization of GFP-CaM-CVIL in the Arabidopsis *ggb* background. On the contrary, in geranylgeraniol (GGol, 20 μM)-treated seedlings, the labeling remains in the nuclei (Figure 4). 

This implies that in the presence of Fol, PFT, which is the only type-I active enzyme in an Arabidopsis *ggb* background, recognizes GFP-CaM-CVIL as a protein substrate. It has been proposed that Fol is not directly used as a substrate but induces the production of MVA by activating HMGR [37,38]. To explore the potential role of MVA in modifying PFT properties under stress conditions, we examined whether MVA production had been promoted under our experimental conditions. We therefore quantified phytosterols, the main MVA-derived metabolites produced in tobacco cells [44]. Cells were treated as described in Figure 1b, and a total sterol fraction (saponified fraction) has been isolated and quantified by GC-FID (Figure 5a). Surprisingly, no significant increase in sterol production could be observed. De facto, a slight drop in sterol levels is observed in the presence of MeJA. To understand this incoherent pattern, we quantified the HMGR levels expressed in these cells. HMGR protein quantification was realized by Western Blotting using microsomal fractions isolated from cells grown as described in Figure 1b and an antibody raised against the catalytical domain of Nt-HMGR2 [39]. Again, in 24 hour-old cells, regardless of treatment, the level of HMGR was not significantly different (Figure 5b). We surmised that levels of HMGR are already high and tested whether subculturing one-week-old cells into new MS-medium could possibly induce HMGR production and mask the likely effects of MeJA (Figure 5c). Interestingly, HMGR levels are high after 24 h and gradually decrease over time, consistent with the quantification of enzyme activity realized previously [27]. Results described in Figure 1a–c suggest that MeJA does not stimulate the accumulation of HMGR protein in tobacco BY-2 cells and does not promote an MVA-derived accumulation of phytosterols. That would mean that the signal we are trying to identify could be independent of HMGR protein accumulation.

If MeJA does not induce HMGR accumulation, we can hypothesize that MVA would be more efficiently incorporated into prenylated protein in the presence of the phytohormone. Thus, we examined whether, after treatment with MeJA, radiolabeled-MVA is more efficiently incorporated into proteins separated (Figure 5d). The corresponding fluorogram showed a slight increase in the radiolabeled signal. Most interestingly, MVA restored membrane localization in *ggb CVIL* plants, similar to what we observed after treating seedlings with MeJA (Figure 5e). Supplementation with DX, a precursor of the MEP pathway, induces membrane localization to some extent (Figure 5e). However, the incorporation of DX into the MEP pathway depends upon the activity of a xylulose kinase catalyzing the formation of DXP [45]. Thus, comparing the effects of MVA and DX can be misleading. Nevertheless, according to these observations, an increase in MVA concentrations can promote the reactivation of protein prenylation in *ggb* mutant lines and modify PFT selectivity in favor of the GFP-CaM-CVIL protein. 

To support the idea that MVA is a link in the reactivation of prenylation following MeJA treatments, we co-treated tobacco BY-2 cells with MeJA with mevinolin (MV, 10 μM), a competitive inhibitor of HMGR, which blocks MVA production. Nuclear localization was recovered in comparable proportions to those observed after treatment with fosmidomycin alone (Figure 6a). This result corroborates the assumption that an MVA-derived precursor has been used. Given that MV is unable to inhibit prenylation of GFP-CaM-CVIL (Appendix A), this suggests the necessity of turning off the MEP pathway (Figure 6a). Finally, we tested if MVA could act in synergy with MeJA in re-establishing membrane localization of fosmidomycin/mevinolin-treated tobacco BY-2 cells (Figure 6b). Before inducing protein expression, we blocked both isoprenoid biosynthesis pathways to ensure that we prevented all endogenous substrate supplies. A concentration range of MVA from 50 to 1000 μM has been tested in the presence (+) or absence (−) of MeJA (20 μM). Up to 200 μM, MeJA acts as a potentiator with MVA. At MVA concentrations ≥500 μM, MVA alone is capable of reverting the isoprenoid biosynthesis inhibition (Figure 6b). Overall, our observations suggest that MVA might be a trigger for altering the substrate specificity of PFT by MeJA.

## 3. Discussion

### 3.1. Modulation of PFT Substrate Specificities by MeJA

We found that a MeJA-induced signal modulates the protein substrate selectivity of PFT, enabling MVA-dependent prenylation of a protein substrate typically associated with PGGT activity and an MEP-derived co-substrate. This implies that specific physiological conditions requiring well-defined signals can modulate and extend CaaX motif recognition by PFT. However, at this point, it cannot be excluded that a different prenyltransferase may be activated to catalyze the modification. Indeed, other types of protein prenylation have been described in eukaryotes. A second class of enzymes (PGGT-II; EC 2.5.1.60) catalyzes the modification of RAB proteins on two C-terminus cysteines using two geranylgeranyl groups [2]. Recently, a third class of enzymes has been described in animals. It is proposed that this family catalyzes the transfer of a geranylgeranyl moiety onto previously farnesylated proteins such as YKT6 (UniProtKB-O15498) or FBXL2 (UniProtKB-Q9UKC9), both containing an additional cysteine generating a CCaaX motif [46,47]. However, this option seems unlikely since both enzyme types are specific to a motif containing two cysteines.

MeJA belongs to the jasmonate family of compounds, which are linolenic acid-derived phytohormones acting as elicitors and inducing defense responses [48]. These phytohormones impact the activity of different kinds of enzymes, mainly by stimulating their gene expression [49,50]. MeJA is known to induce the production of various metabolites, including isoprenoid compounds. This is how triterpene saponins are induced in *Medicago truncatula* [51], the sesquiterpene artemisinin in *Artemisia annua* L. [52], or monoterpenes in *Picea glauca* [53]. Concomitantly, specialized metabolism is activated, while ROS production can be circumscribed and growth controlled [54,55,56]. Furthermore, in *Catharanthus roseus*, the production of monoterpene indole alkaloids (MIA) induced by the jasmonate signaling pathway is controlled by protein geranylgeranylation [57,58], with a paramount contribution from the plastidial CrGGPPS2 [59]. In a similar way, MEP-derived protein prenylation is mandatory for the production of MVA-derived sesquiterpenoid phytoalexins after the elicitation of tobacco plants with cellulase [39]. Collectively, these studies demonstrate a clear correlation between the induction of the production of specialized metabolites required for defense responses and protein prenylation in plants.

Although knowledge remains limited, the regulation of eukaryotic protein prenylation is better characterized in humans, animals, and yeast than in plants [21]. Yet, it has been described that, on a molecular level, a tryptophan at position 110 should be important for selectivity against GGPP (https://www.uniprot.org/uniprotkb/Q38920/entry (accessed on 1 September 2023)). Since PPTs have to process a multitude of substrates, it is essential that catalytical properties are precisely controlled within the cell. It is accepted that in vitro, the recombinant Arabidopsis PFT recognizes GFP-CaM-CVIL, the protein substrate known as a PGGT-specific substrate, in similar affinity ranges as its dedicated substrate (GFP-CaM-CVIM), but the turn-over is seven times lower [13]. Regulation of enzyme activities in planta, has so far not been characterized. Here, we demonstrated that GFP-CaM-CVIL stays unmodified when expressed in the *ggb* mutant line, where only PFT is active. Thus, PFT appears to be unable to modify GFP-CaM-CVIL under standard growth conditions in Arabidopsis and shows higher substrate selectivity in planta than in vitro. However, the capacity of PFT to modify GFP-CaM-CVIL can apparently be initiated following MeJA treatment. Therefore, it is possible that the microenvironment of enzymes is highly dynamic and requires constant readjustment. At this point, it can be proposed that MeJA modulates the cellular environment, controlling PFT activity and substrate specificity. Jasmonate-mediated signal responses are known to involve and modulate post-translational modifications (PTMs) [60]. In this context, we can mention phosphorylation in the common α-subunit, which increases enzyme activity [21]. For example, there is evidence that human enzyme activity can be enhanced by insulin following the phosphorylation of serine residues in the α-subunit [21]. Our case study does not provide clear indications that PFT activity has been stimulated, but rather that substrate recognition has been modified. It is unknown whether PTMs can modulate substrate specificity, and no investigations have been carried out to date. Interestingly, our study pointed to two ways to regulate PFT selectivity: either by blocking GGPP biosynthesis through the action of fosmidomycin or by inactivating PGGT in the Arabidopsis *ggb* mutant. Both conditions are prerequisites for MeJA activity in PFT regulation.

### 3.2. MVA as a Sensor for Cross-Talk between MVA and MEP Pathway?

This study highlighted the possibility of MVA being a central metabolite linking MeJA signaling to the regulation of PFT activity in plants. Such a strategy would imply that MVA production operates as a sensor, triggering metabolic cross-talk by mandating the utilization of an MVA-derived metabolite instead of a MEP-derived prenyl diphosphate to modify a CaaL-motif protein.

The role of MVA is of central importance in stimulating the biosynthesis of isoprenoid compounds [61], as well as supposedly MEP-derived metabolites [22,34,62,63]. MVA has functions beyond isoprenoid biosynthesis, like sugar and amino acid uptake in human cells [64]. Regulation of the MVA biosynthesis pathway is also related to energy sensors. For instance, extracellular ATP activates the metabolism [65], while the energy sensor AKIN10-GRIK1 inhibits HMGR via protein phosphorylation [66]. MVA synthesis can be rapidly promoted by stimulating HMGR activity, often through protein overproduction. Up to ten different HMGR isoforms responsible for MVA biosynthesis coexist in a plant [67], most of them with specific functions and different spatio-temporal patterns. We failed to identify a specific isoform involved in the regulation of protein prenylation. However, it has to be mentioned here that an increase in a particular HMGR isoform, which would be devoted to protein prenylation, may not be quantifiable through Western Blotting when employing an antibody that recognizes all forms. The existence of MVA metabolons being carefully regulated has been proposed [68,69]. Those dynamic supramolecular complexes would drive specific MVA-derived metabolite production under particular conditions through substrate channeling. However, it cannot be excluded at this point that such functions are assigned to an MVA-derived metabolite. Nevertheless, MVA-induced changes in the profile of small GTP-binding proteins have been described [70]. In the same molecular range as those small GTP-binding proteins, we observed a slight increase in MVA incorporation in MeJA-treated cells. These proteins are notably involved in defense signal-transduction pathways [71], but not all of them are prenylated. In this context, the importance of membrane localization of regulatory elements in phytohormone signaling pathways has been reported. This is, for instance, the case of myristoylated calcium-dependent protein kinases [72]. Protein prenylation is also critical upstream of jasmonate signaling, as VIGS of the PGGT β-subunit in Catharanthus cells and plants negatively impacts the expression of MeJA-induced transcription factors [58,59]. More recently, it has been proposed that some Catharanthus Rho of Plant (ROP) proteins act as transcriptional activators within the nucleus [73]. The question of whether MVA can be sensed in the context of protein prenylation regulation has not been described to date, but it is a conceivable possibility.

### 3.3. A Working Model of How Plants Benefit from Modifying PFT Substrate Selectivity

The utilization of MVA-derived prenyl diphosphates, in the presence of MeJA, to modify GFP-CaM-CVIL sensor proteins raises a number of interrogations. One question at hand is whether it is beneficial to adjust the substrate specificity of a PPT. It is likely that the importance of protein prenylation in plant stress responses [2] is part of the answer. Since enzyme-catalyzed reactions are limited by substrate supply, such a strategy involving two enzymes and two metabolic pathways could promote biocatalysis more effectively by commandeering substrates from a wider range of sources. Thus, using a substrate generated by the MVA pathway, rather than or in addition to the MEP pathway, to modify a CaaL-motif protein important for the stress response would enable this to be performed more rapidly.

We propose a model for prenylation of GFP-CaaL proteins in which a stress-induced signal modifies the substrate specificity of PFT (Figure 7). In unstressed tissues, PGGT is the active enzyme responsible for modifying CaaL-like proteins using a prenyl diphosphate group (GGPP) derived from the plastidial MEP pathway. This assessment is substantiated by the nuclear distribution of GFP-CaM-CVIL expressed in Arabidopsis *ggb* plants and the loss of membrane localization subsequent to fosmidomycin treatments.

The stress-induced signal might be correlated with an increase in the amount of MVA or an MVA-derived metabolite. It is anticipated that under stress conditions, PPTs will become less stringent toward dedicated protein substrates, which would increase their capacity to modify proteins needed in signaling pathways supporting stress responses. Abiotic stresses related to bites or wounding induce some signals that could potentially lead to MeJA production that will modify PFT substrate specificity. As a result, both PFT and PGGT are modifying the same protein, causing an increase in the prenylation capacity and potentially in the way the protein is modified (FPP vs. GGPP).

## 4. Materials and Methods

### 4.1. Chemical Materials 

All-*trans*-farnesol, all-*trans*-geranylgeraniol, farnesyl diphosphate, geranylgeranyl diphosphate, dexamethasone, fosmidomycin, methyl-jasmonate, ethephon and mevalonolactone were purchased from Sigma-Aldrich (Saint Quentin Fallavier, France). Cellulase RS was obtained from Yakult Pharmaceutical Industry (Tokyo, Japan), and mevinolin was a kind gift from Drs. M. Greenspan and A.W. Alberts (Merck Sharp and Dohme, Rahway, NJ, USA). Before use, the lactone forms of mevinolin and mevalonolactone were converted into their free salt forms, as described by [28]. 1-deoxy-d-xylulose (DX) was obtained from AlsaChim (Illkirch Graffenstaden, France). (*R*,*S*)-[2-^14^C]mevalonolactone was purchased from Amersham. Analytical-grade solvents were purchased from Sigma-Aldrich (Saint Quentin Fallavier, France), Carlo Erba (Val-de-Reuil, France), or Fischer Scientific France (Illkirch-Graffenstaden).

### 4.2. DNA Constructs, Biological Materials and Culture Conditions 

Two sensors for protein prenylation have been used in this study: GFP-CaM-CVIL, a GFP protein fusion with the C-terminus side of rice Calmoduline 61 (GenBank: AAA98933) and GFP-ROP6-CSIL, a fusion protein of GFP with Rho of Plant ROP6 (GenBank: OAP00270). Both coding sequences have been cloned into pTA7001 and are under the control of a dexamethasone-inducible promoter [74]. The pTA-GFP-CaM-CVIL construct was generated as previously indicated [28]. To construct pTA-GFP-ROP6-CSIL, *rop6* has been amplified by PCR using ROP6-F 5′-GCAGAGCTCAGTGCTTCAAGGTTTATCAAG-3′ and ROP6-R 5′-TCTTCTAGATTCAGAGTATAGAACAACCTTTCTGAGATTTTCTC-3′ as a primer set and using as a template, cDNA prepared from Arabidopsis seedlings [45]. The fragment was cloned between *SacI* and *XbaI* restriction sites in the frame of the *GFP* of the pMRC-GFP vector [75]. The *XhoI* and *Spe*I restriction sites were incorporated by PCR using the primer sets XhoIGFP-*F* 5′-CCGCTCGAGGGATGGTGAGCAAGGGCGAGGAGC-3′ and SpeIROP6-R GGACTAGTCCTCAGAGTATAGAACAACCTTTC-3′ to subclone *GFP-ROP6-CSIL* into pTA7001. 

Suspension-cultured tobacco (*Nicotiana tabacum*) cv. BY-2 cells were maintained in modified Murashige and Skoog medium as previously described [76]. The generation of transgenic tobacco BY-2 cells expressing GFP-CaaL proteins under the control of the inducible dexamethasone promoter was obtained by co-culturing 10-day-old calli with pTA-GFP-CaM-CVIL or pTA-GFP-ROP6-CSIL-transformed *Agrobacterium tumefaciens* LBA4404.pBBR1MCS-5.virGN54D strain [77]. After a selection of hygromycin-resistant calli (30 μM), a homogenous cell suspension was regenerated. Protein expression is induced by 15 μM of dexamethasone. 

All *Arabidopsis thaliana* mutants and transgenic lines are in the Columbia-0 (Col-0) ecotype background. After ethanol and bleach-sterilization, seeds were set to germinate on half MES-Murashige and Skoog medium supplemented with 0.8% agar. Transformed Col-0 plants (*Col-0 GFP-CaM-CVIL*) and T-DNA loss of function mutant *ggb-2* plants [7] (*ggb GFP-CaM-CVIL*) expressing GFP-CaM-CVIL were obtained from Prof. D.N. Crowell (Idaho State University, Pocatello, ID, USA). Lines have been transformed by floral dip with pTA-GFP-CaM-CVIL-transformed *Agrobacterium tumefaciens*. The T-DNA insertion farnesyltransferase β-subunit loss of function *era1-9* mutant was obtained from the Arabidopsis Biological Resource Center under the reference SALK_110517 (https://www.arabidopsis.org/servlets/TairObject?type=polyallele&id=500194245 (accessed on 25 May 2013)). Because the transformation by floral dip was inefficient, *era1.9 pTA-GFP-CaM-CVIL* was obtained by crossing era1.9 with WT pTA-GFP-CaM-CVIL. For transformation of *era1.9*, Col-0 GFP-CaM-CVIL and *era1.9* seedlings were transferred to soil and grown under short-day conditions (16 h light (22 °C)/8 h dark (18 °C) photoperiod). Father *era1.9* plants were crossed with Mother *Col-0 GFP-CaM-CVIL* plants and hygromycin (30 μM)-resistant plants were screened for GFP expression after induction with dexamethasone (30 μM). Transformed homozygote *era1* plants were selected by PCR genotyping using *era5F* 5′-ACCTACTGTGGTTTGGCTGC-3′, *era5R* 5′-CAACAACGGGTCATGCTGCT-3′ and *Lb1b* 5′-TGGCAGGATATATTGTGGTG -3′. 

Tobacco (*Nicotiana tabacum*) wild-type (var. xanthi line SH6) was grown as an axenic shoot culture on hormone-free MS-medium (Duchefa, The Netherlands). Transformation was conducted by co-cultivation of *Agrobacterium tumefaciens* GV3101 harboring the construct pTA-GFP-CaM-CVIL and 5–6-week-old leaf disks [78]. Following dexamethasone induction (30 μM), three independent-hygromycin-resistant (30 μM) transformed F1 generation plants were selected by confocal microscopy, followed by subsequent analyses for fluorescence emission. 

### 4.3. Microscopy Analyses

For tobacco microscopy analyses, GFP-CaaL proteins are expressed by adding dexamethasone (15 μM for tobacco cells, 30 μM for seedlings, during 18 h) to biological material pretreated with chemicals (3 h for tobacco cells, 6 h for seedlings). This step is necessary to observe GFP-CaaL prenylation statues according to isoprenoid pool availabilities. Tobacco leaf disks (1 cm in diameter) were punched from 5–6-week-old plants and positioned adaxial side upward onto MS-medium water and treated as indicated in Figure 3. Axenic Arabidopsis 2-leaf seedlings were transferred to MS-medium and treated for 24 h in 12-well plates before protein expression was induced by adding 200 μL of MS-medium supplemented with 30 μM dexamethasone. Tobacco BY-2 cells (>300) were analyzed in two independent measurements and classified by visual selection into the following categories: plasma membrane-localized, nucleus-localized, or both. The efficiency of cellular-treatments with inhibitors was evaluated using chi-squared tests against the null hypothesis being true. Significant differences were assigned to *p*-values < 0.01. Acquisitions of images were realized with a Zeiss LSM700 confocal microscope equipped with a Plan-Apochromat 20×/0.8 M27 objective using a 488 nm laser (15%) for excitation. Fluorescence emission was revealed by using the SP555 filter and the Main Dichroic beam splitter MBS 405/488/555/639. Acquisitions were handled using ZEN 2009 software (Carl Zeiss, Tokyo, Japan; http://www.zeiss.com/ (accessed on 12 October 2020)), exported as Tagged Image Files, and processed with Adobe Photoshop 5.0 software (Adobe Systems).

### 4.4. Phytosterol Analyses

Phytosterol extraction was conducted as previously described by [79] using 100 mg (dry weight) of tobacco BY-2 cells. Experiences were performed in triplicate, and sterols were quantified using α-cholestane as a standard and calculated as a percentage, taking the mean of total phytosterol content in control cells as a reference. 

### 4.5. Protein Quantification, Assays and Fluorography

Microsomal protein fractions were prepared as previously described [76]. Protein concentration was quantified by a Bradford method using BSA as a standard and the manufacturer’s protocol (BioRad, Marnes-la-Coquette, France). HMGR contained in 40 μg was revealed and quantified by Western blotting using a polyclonal rabbit antibody raised against the soluble domain of *N. tabacum* HMGR2 [39]. For the fluorogram, tobacco BY-2 cells (500 μL of 1 week-old cells diluted 5-times in MS-medium) were incubated with 1 μCi of [^14^C]MVA for 24 h at 22 °C on a tube rotator. Cells were recovered by centrifugation and washed twice with PBS buffer. Proteins were extracted with 200 μL TCA (10%) and precipitated by adding 800 μL cold acetone 80%. The pellet obtained was dried under vacuum and recovered in 75 μL of Laemmli loading buffer. A 12% SDS-PAGE gel was loaded with the mixture. The fixed gel was activated with Amplify from Amersham and exposed to a preflashed X-ray film for 1 month before being scanned and processed with Adobe Photoshop 5.0 software (Adobe Systems). 

## 5. Conclusions

This study demonstrated that PPT substrate specificity is modulable under phytohormonal orchestration in planta by a so-far unknown mechanism. We provided evidence that MeJA rearranges the ability of PFT, an enzyme, known to transfer farnesyl moieties, to specifically use a protein substrate typically dedicated to its sister enzyme PGGT-I. These results suggest that prenylated proteins might possibly be modified by different prenyl groups depending on the physiological conditions and plant growth. The possibility to regulate PPT activities and substrate specificities in planta offers a way to interplay with the use of two pathways for prenyl diphosphate biosynthesis, with MVA conceivably acting as a sensor metabolite underpinning this dialog. In addition, this pool-driven flexibility of enzyme substrate specificity may be essential in plant stress responses. MeJA is an important inducer employed to produce valuable secondary metabolites in plants [80]. Understanding the regulation of protein prenylation will provide a better understanding of plants’ ability to produce valuable specialized metabolites and identify new strategies for their accumulation, for instance, by using both isoprenoid biosynthesis pathways.

## Figures and Tables

**Figure 1 plants-13-01110-f001:**
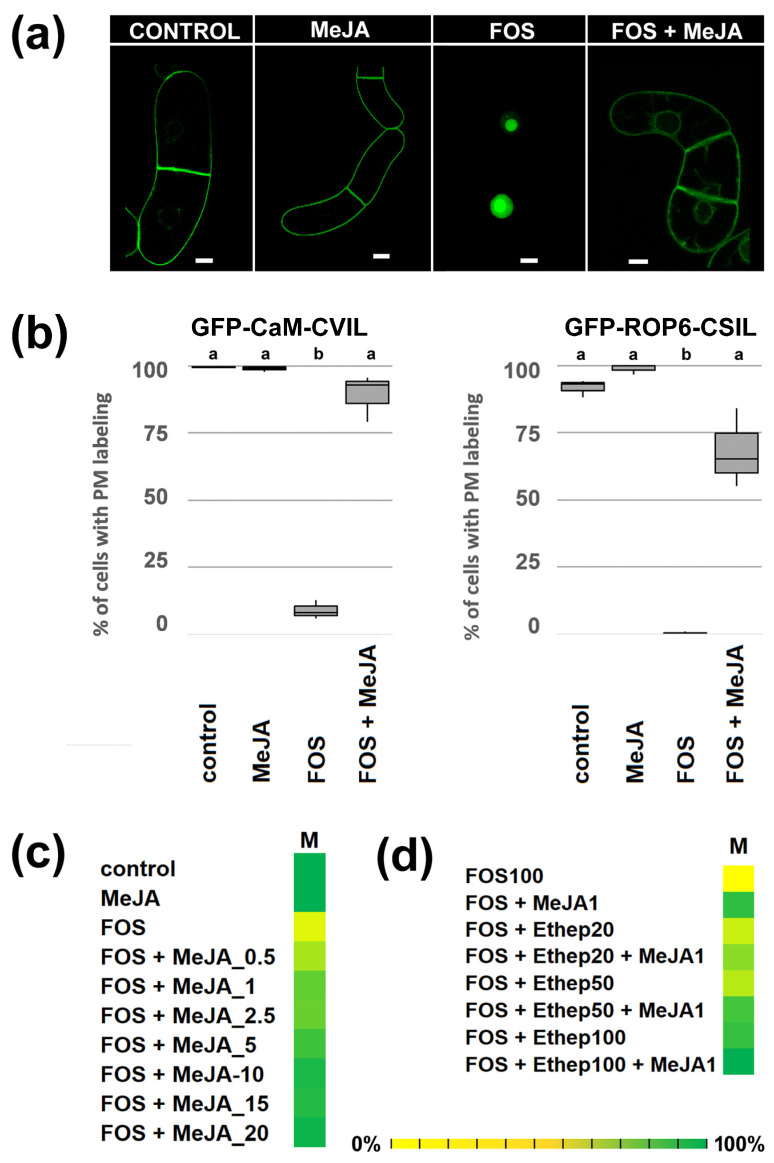
Jasmonic acid methyl esther (MeJA) restores inhibition of GFP-CaaL protein prenylation by fosmidomycin (FOS). (**a**) Confocal pictures of control, fosmidomycin/FOS (100 μM) and FOS/MeJA (100 μM/20 μM) cells expressing GFP-CaM-CVIL under the control of the dexamethasone-inducible promoter. Seven-day-old cells (stationary phase) were diluted 5 times in new MS medium and cultivated for 3 h in the presence of inhibitors before induction with 30 μM dexamethasone. (**b**) Cells expressing GFP-CaM-CVIL have been categorized into 2 classes: cells without a nucleus localization and cells in which nuclei are labelled. Three independent sets of experiments were performed, in which at least 300 localizations were evaluated. A Box and Whisker plot was used to illustrate the results. Groups with different letters have significantly different means (*p* ≤ 0.005) as calculated by a Bonferroni–Holm test following a one-way ANOVA. (**c**) Heat maps showing the dose-dependent effect of MeJA and the additive effect of ethephon (**d**). The color scale from yellow (0%) to green (100%) indicates the percentage of cells with membrane localization. The more the color is green, the better membrane localization is effective. Cells with membrane localizations (M) were counted (*n* > 300).

**Figure 2 plants-13-01110-f002:**
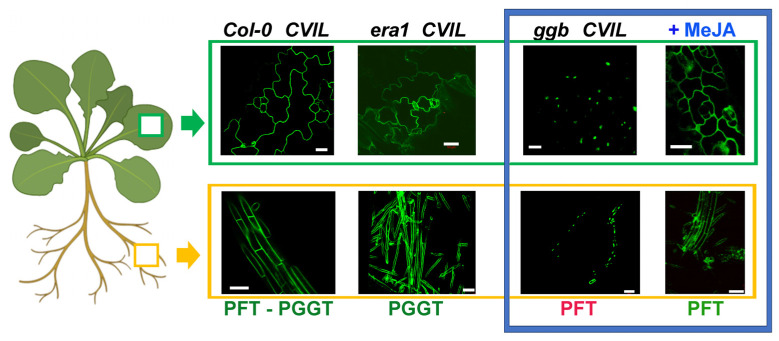
Representative confocal pictures depicting subcellular localization of GFP-CaM-CVIL proteins expressed in Arabidopsis lines and activation of protein prenylation by MeJA in *ggb* lines. Col-0, *era1*, and *ggb* lines have been transformed with pTA-GFP-CaM-CVIL constructs (CVIL), and leaves (green scare) or roots (yellow scare) have been observed by confocal microscopy. The WT Arabidopsis background retains both PFT and PGGT activities, while in loss of function *era1* lines, only PGGT-I is expressed, and in *ggb* lines, only PFT remains active. In *ggb CVIL* lines, the protein cannot be modified, indicating that PFT alone cannot modify GFP-CaM-CVIL under standard growth conditions (red letters, left side in the blue rectangle). In the presence of MeJA (in blue), the membrane localization of the GFP-modified protein is again observed in *ggb* plants, suggesting an active PFT (green letters (right side in the blue rectangle). The white bar corresponds to 50 μm. On the left, the Arabidopsis plantlet has been created using biorender (https://www.biorender.com/ (accessed on 15 October 2023)).

**Figure 3 plants-13-01110-f003:**
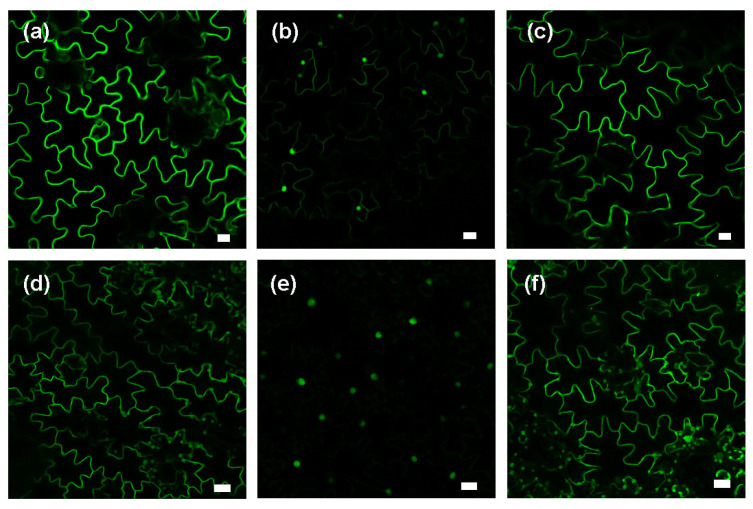
Treatment with jasmonic acid methyl esther (MeJA) increases the biosynthetic capacity of protein prenylation in *Nicotiana tabacum* leaf disks. (**a**,**d**) control; (**b**,**e**) fosmidomycin (100 μM) and mevinolin (5 μM) treatments. (**c**) represents condition (**b**) in the presence of 0.5% cellulase and (**f**) condition (**e**) in the presence of 30 μM MeJA. Leaf disks were pretreated for 6 h with chemicals as indicated below before GFP-CaM-CVIL expression was induced with 30 μM dexamethasone. During the whole experiment, plant materials were incubated in the dark at 22 °C. The white bar represents 20 μm.

**Figure 4 plants-13-01110-f004:**
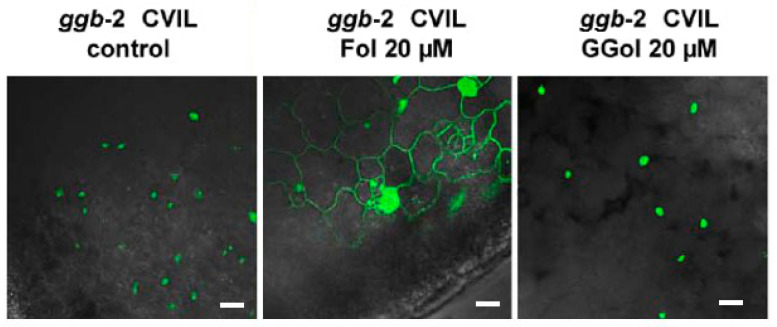
Farnesol (Fol)-induced protein prenylation in leaves of Arabidopsis *ggb CVIL* mutant plants transformed with pTA-GFP-CaM-CVIL. The reversion does not work with geranylgeraniol (GGol). Subcellular localization of GFP-CaM-CVIL was evaluated by confocal microscopy. The white bar corresponds to 20 μm.

**Figure 5 plants-13-01110-f005:**
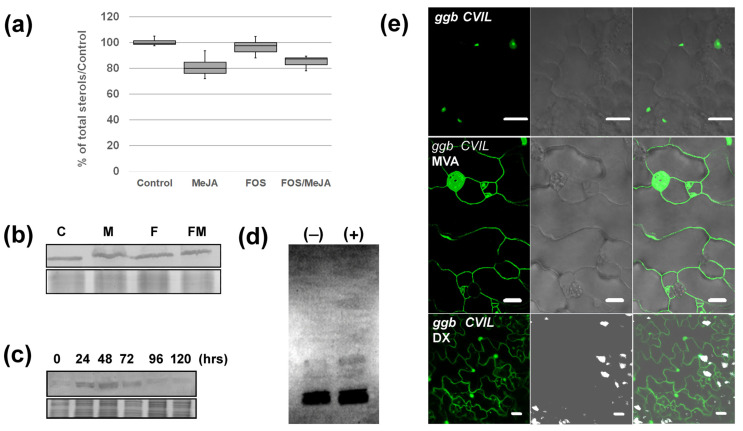
Mevalonic acid (MVA) modifies PFT protein substrate specificity. (**a**) Quantification of total phytosterol isolated from untreated cells (C), cells treated with MeJA (30 μM), with fosmidomycin (FOS 100 μM) and MeJA/FOS have been deposited in each well. Groups have no significantly different means (*p* ≥ 0.005) as calculated by a Bonferroni–Holm test following a one-way ANOVA. (**b**) Quantification of HMGR protein by Western Blotting using a polyclonal antibody raised against the catalytical domain of tobacco HMGR2. A total of 40 μg of a microsomal protein fraction from tobacco BY-2 cells treated as described in (**a**) have been charged in each well. (**c**) Quantification of HMGR in tobacco BY-2 cells during growth curve. Cells were diluted 5-fold in MS medium and grown for 24-h, 48 h, 72 h, 96 h and 120 h before being collected to make microsomal fractions. (**d**) [^14^C]MVA (1 μCi) incorporation into prenylated proteins of tobacco BY-2 cells treated (+) or not (−) with 20 μM MeJA during 24 h. The picture represents a fluorogram of SDS-PAGE-separated total protein extracts. (**e**) Subcellular localization of GFP-CaM-CVIL protein expressed in Arabidopsis *ggb* lines grown for 48 h in the presence of MVA (2 mM) or DX (2 mM) before protein expression is induced with dexamethasone (30 μM).

**Figure 6 plants-13-01110-f006:**
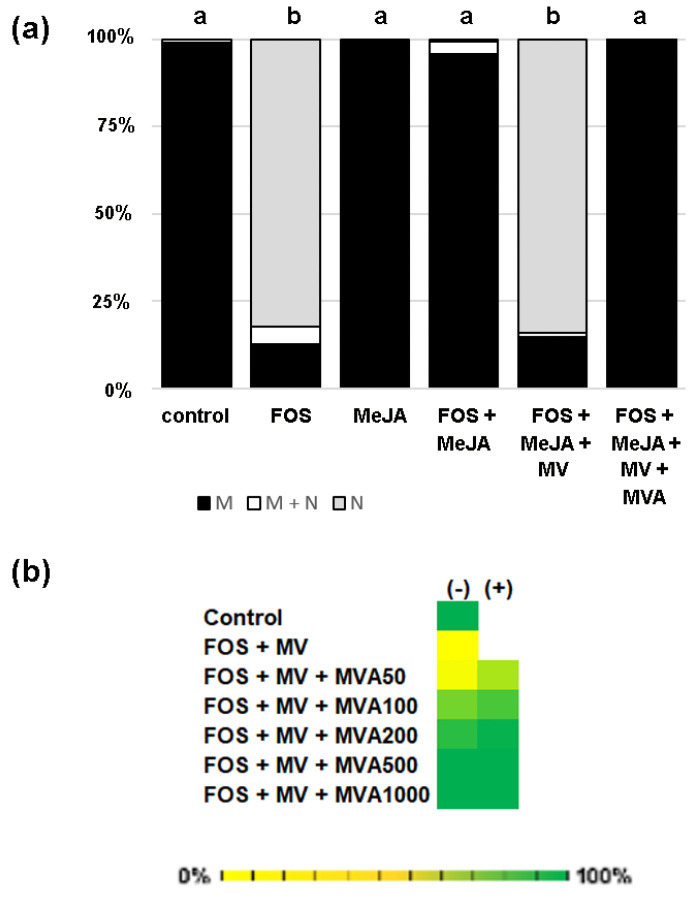
Mevalonic acid (MVA) as a candidate for providing prenyl groups in MeJA/FOS-treated tobacco BY-2 cells. In each experiment, cells were pretreated for 3 h with chemicals as indicated, before protein expression was induced with dexamethasone (15 μM). (**a**) Mevinolin (MV 10 μM) inhibits MeJA (20 μM)-mediated reversion of FOS (100 μM). Treated with FOS + MeJA + MV, the nuclear localization of GFP-CaM-CVIL expressed in tobacco BY-2 cells is revived. MVA (2 mM) reverses MV-induced inhibition of protein prenylation in the presence of FOS and MeJA. Distribution of GFP fluorescence in membrane (M, black) membrane/nucleus (M + N, white) and nucleus (N gray) is indicated in percentage and chi-squared tests against the null hypothesis being true were used to assign significant differences to *p*-values < 0.01, which are indicated by different letters. (**b**) Synergetic effect of increasing concentrations of MVA with (+) or without (−) MeJA (20 μM) in FOS/MV (100 μM/5 μM)-treated tobacco BY-2 cells. Cells with membrane localizations (M) were counted (*n* > 300). Color scale from yellow (0%) to green (100%) indicates the percentage of cells with membrane localization. The more the color is green, the better membrane localization is effective.

**Figure 7 plants-13-01110-f007:**
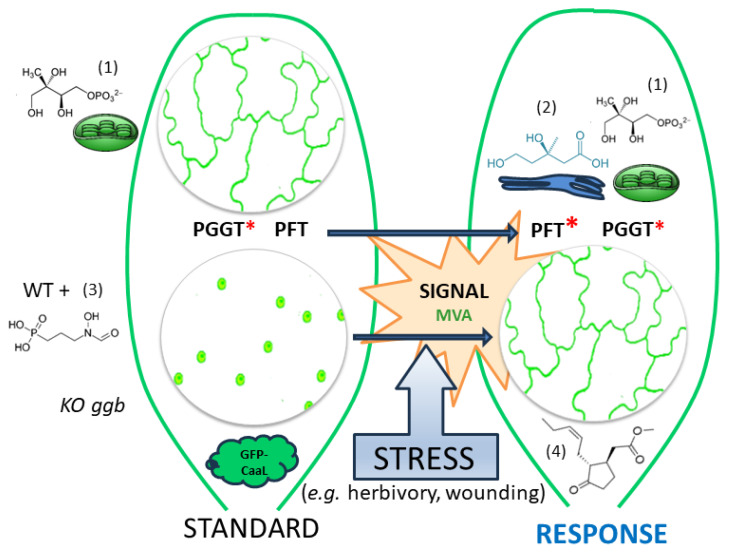
A working model of the role of jasmonic acid methyl esther (MeJA) in the activation of protein prenylation through isoprenoid metabolic cross-talk. Under standard growth conditions, the protein is modified with a plastidial methylerythritol phosphate (1) pathway-derived geranylgeranyl group and is labeling the plasma membrane [28]. Unprenylated, the protein localizes in the cell nucleus. The absence of prenylation is attributed to the inhibition of the MEP pathway by fosmidomycin (3) or the absence of PGGT in the Arabidopsis *ggb* KO mutant. Treatment with MeJA (4) leads to a signal stimulating the use of mevalonic acid (MVA) (2)-derived substrates to improve the capacity for protein prenylation in plants under stress conditions. This signal modifies the substrate specificity of PFT, which recognizes GFP-CaM-CVIL under those conditions. The increase in prenylation capacity allows a reversion of fosmidomycin inhibition and the capacity of PFT to recognize a PGGT protein substrate. The red * indicates which enzyme is modifying GFP-CaaL proteins.

## Data Availability

The data presented in this study are available on request from the corresponding author.

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
