# Peer review of "Methyl-Jasmonate Functions as a Molecular Switch Promoting Cross-Talk between Pathways for the Biosynthesis of Isoprenoid Backbones Used to Modify Proteins in Plants"

_plants, 2024, doi:10.3390/plants13081110_

Round 1

Reviewer 1 Report

Comments and Suggestions for Authors

The authors provide some evidence for a mechanism by which Me-JA affects protein prenylation. However, the manuscript still needs some minor revisions before it can be accepted.

1) The first sentence of the abstract mentions GFP-CaaL without any explanation, which is not friendly to a larger readers to understand the manuscript. Also, what is the last sentence of the abstract trying to convey? It seems quite unnecessary.

2) In Figure 1, the picture of MeJA alone (Figure 1a) is missing; please add the abbreviation of fosmidomycin;

3) Line 165, please add the full name of NLS;

4) Line 184, why the authors choose phytohormones for their study? The authors did not mention any reason in the Introduction;

5) ‘Arabidopsis thaliana’, ‘Arabidopsis’, ‘Arabidopsis’, Please select one and keep it consistent throughout the manuscript;

6) ‘Jasmonic acid’ or ‘JA’? Choose one and keep it consistent throughtout the manuscript.  

Author Response

The authors provide some evidence for a mechanism by which Me-JA affects protein prenylation. However, the manuscript still needs some minor revisions before it can be accepted.

1) The first sentence of the abstract mentions GFP-CaaL without any explanation, which is not friendly to a larger readers to understand the manuscript. Also, what is the last sentence of the abstract trying to convey? It seems quite unnecessary.

Thank you for raising these concerns.

The abstract has accordingly been modified and the last sentence has been deleted.

2) In Figure 1, the picture of MeJA alone (Figure 1a) is missing; please add the abbreviation of fosmidomycin;

A picture of MeJA-treated tobacco cells has been added and the abbreviation of fosmidomycin added in the legend.

3) Line 165, please add the full name of NLS;

Nuclear localization sequence (NLS) has been added

4) Line 184, why the authors choose phytohormones for their study? The authors did not mention any reason in the Introduction;

Thanks for highlighting this point. We actually did not choose. We screened over 110 conditions and identified MeJA as a positive hit, which was further investigated in this study. Of course, phytohormonal treatments were part of this screening. To avoid any misunderstanding, at the end of the introduction, we deleted “with a special emphasis given to hormonal regulation”.

5) ‘Arabidopsis thaliana’, ‘Arabidopsis’, ‘Arabidopsis’, Please

select one and keep it consistent throughout the manuscript;

It has been homogenized in the manuscript.

6) ‘Jasmonic acid’ or ‘JA’? Choose one and keep it consistent

throughtout the manuscript.

It has been homogenized in the manuscript.

Reviewer 2 Report

Comments and Suggestions for Authors

MVA did act as a trigger for restoration of membrane location, but how about the ndogenous MVA increased in the cells treated with Me-JA? How Me-JA alters the substrate speciffcity of PFT still remains unclear, the direct check of protein prenylation might be necessary in the situation of inhibition restoration besides location experiment, and the prenyl groups for the prenylated proteins also need be noted (C20 orC15?).

 Line340, the P value dose not properly support the statement ‘no significance’.

There is no white bar as noted on any picture in Fig4.

Comments on the Quality of English Language

There are some errors in the context, e.g.  Line241, path7ways.

Author Response

MVA did act as a trigger for restoration of membrane location, but how about the ndogenous MVA increased in the cells treated with Me-JA? How Me-JA alters the substrate speciffcity of PFT still remains unclear, the direct check of protein prenylation might be necessary in the situation of inhibition restoration besides location experiment, and the prenyl groups for the prenylated

proteins also need be noted (C20 orC15?).

Thank you for this constructive suggestion. We totally agree. In fact we are trying since many years to address this, until now, unsolved problem. The analysis and identification of prenyl modifications is very tricky for several reasons.  1. How to be sure that a farnesyl of a geranylgeranyl group was linked to a protein and not just interact with this protein. 2. Most of the proteins are quite hydrophilic, the isoprenyl group is hydrophobic. Very often, prenylated peptides are just lost and analyses are not reproducible: sometimes you get it, sometime not. 3. In addition, plants are rich in many different isoprenyls making the analysis even more difficult.

Most of the techniques used to analyze prenyl groups do not consider these points. There are techniques tagging proteins via alkyne-modified prenyl substrate feeding, but these techniques do no consider regulations like those we propose in this manuscript.

For all these reasons, this point cannot be addressed here, but we are convinced that solutions will be proposed in the near future.

Line340, the P value dose not properly support the statement ‘no significance’..

There is no white bar as noted on any picture in Fig4.

Thank you for these remarks, white bars have been added and the sign for p-values was replaced by ≥.

Comments on the Quality of English Language There are some errors in the context, e.g. Line241, ‘path7ways’.

The whole manuscript has been reedited.

Reviewer 3 Report

Comments and Suggestions for Authors

In “Methyl-jasmonate functions as a molecular switch promoting cross-talk between pathways for the biosynthesis of isoprenoid backbones used to modify proteins in plants,” Chevalier et al. observe that methyl jasmonate can overcome the inhibition of protein prenylation by fosmidomycin. Through the use of various chemical inhibitors, chemical inducers, and mutant lines, they propose a model in which jasmonic acid signaling loosens substrate specificity of the enzyme protein farnesyl transferase. An excess of farnesyl diphosphate can also achieve similar results. These results seem to be of interest and worth publishing, but I believe require major revisions for interpretation.

Conceptual/interpretation issues:

·         If I interpret correctly, there are two mechanisms by which the GFP-CVIL gene can be prenylated in the absence of its preferred prenyl partner geranylgeranyl diphosphate (GGPP) or the prenylating enzyme Protein GGPP Transferase (PGT). In both cases, it seems that the protein is prenylated with farnesyl diphosphate (FPP) by Protein Farnesyl Transferase (PFT). In the first mechanism, this happens through an abundance of FPP (Figure 5E,  Figure 6AB, reference 38). In the second, MeJA allows PFT activity despite a lack of increased FPP (Figure 5A, lines 331-335). This distinction is not made clearly in the text. In fact, Figure 3 and discussion of it (lines 265-270) suggest that MeJA may increase FPP levels, which is later argued against. If I interpreted correctly the authors should do some extensive re-writing to make this distinction clear.

·         In lines 300-302, the authors propose that, because radio-labeled FPP is incorporated to GFP-CVIL at a higher level than radio-labeled GGPP, the primary prenylation should be farnesylation via PFT. This is inconsistent with the results that fosmidomycin (GGPP inhibitor) blocks membrane localization more than mevinolin (FPP inhibitor), and with the result that pft mutants are able to correctly localize prenyl-substrate GFP while pgt mutants are not. Those results indication that GGPP should be the important incorporated prenyl, but the GGPP incorporation in figure 4B is so weak that to make conclusions based on a reduction or lack of reduction does not seem well-justified. This conflict should be fully explored, and perhaps the GGPP incorporation assay repeated with a greater level of radiolabeled substrate or higher exposure, to justify any conclusions from either assay.

·         The authors rely on a difference between membrane localization and nuclear localization, but do not explicitly test membrane localization vs cytosolic localization. As the cytosol is generally held to cell periphery by the vacuole, these can be hard to separate by microscopy. Can the authors test that the MeJA-induced ‘membrane localization’ is truly a restoration of prenylation, and not some inhibition of nuclear localization, leading to protein accumulation in membrane-adjacent cytosol?

Data display and visualization issues:

·   I don’t have access to Supplemental Figure 1, so I cannot be sure exactly what it contains, but I would like to know the full list of compounds tested for ability to rescue FOS-induced mislocalization, and their effects. This would help to evaluate the significance of the MeJA result.

·   Figure 1: The red-green color scale in panel B will cause issues for readers with red-green colorblindness. Please replace with a color-blind friendly scale, or better yet show value and distribution as in panel B. Was the experiment in panels C and D replicated? Are there statistical differences among these treatments?

·   Figure 2: was this experiment replicated? Are these results consistently observed?

·   Figure 3: was this experiment replicated? Are these results consistently observed?

·   The determination of FOS100/MV5 is not clear from the text. A supplemental figure showing the concentration combinations tested would help readers to understand the significance of pathway inhibition at these concentrations.

·   Figure 4 panel A microscopy images lack scale bars. For both panels A and B was this experiment replicated? Are these results consistently observed?  For panel B, as stated in conceptual issues, the GGPP bands seem too faint to justify strong conclusions. Is this an issue with exposure? Moreover if the goal is to observe differential prenylation of the substrate, the authors should also present a control for stable levels of GFP protein itself.

·   Figure 6: For panel A, the concentration of MVA used (2 mM) is more than enough to restore membrane localization with or without MeJA according to panel B. A more representative concentration would be better. For panel B, as in Figure 1, the authors should avoid a red-green color scale, and it would be better to show the data itself in the style of panel A or Figure 1. For both panels: was this experiment replicated? Are these results consistently observed?

·   Lines 382-385 indicate that MV does not interfere with membrane localization in the absence of FOS and that this is shown in figure 6A. However, Figure 6A does not include MV or MV+MeJA treatments alone without FOS.

Minor language:

Line 80: in cellula is not a common term, it seems that in vivo would be better in this case

Line 125: HMGR has not been defined as an acronym or given context as a key MVA pathway enzyme.

Line 197,202, etc: Col0 should be Col-0

Line 200: plantlets generally refers to plants regenerating from tissue culture or clonal propagation, while these are Arabidopsis seedlings

Line 241: “path7ways” should be “pathways”

Line 275: ‘Fol’  abbreviation has not yet been defined (farnesol)

Line 312: “donner” should be “donor”

Line 342: @HMGR2cat I believe should be α-HMGR2cat

Line 356: ggb::CVIL – the : or :: are usually used to indicate a promoter and the gene whose expression it drives. This mutant line expressing a protein of interest should be written ggb GFPCVIL or similar

Line 386,390: Synergy refers to a situation where the whole is greater than the sum of the parts. I do not see evidence for synergy necessarily as the combined effect of MeJA and MVA may be additive

Line 477: “VIGS-silencing” VIGS is an acronym for virus-induced gene silencing, the word “silencing” does not need to be repeated

Comments on the Quality of English Language

The paper, while readable and interesting, contains many instances of odd word usage that impede the meaning. An incomplete list of examples includes

·         Line 227: “MeJA endorses protein prenylation capacity in tobacco” To endorse is to give approval. Enhances may be a better choice

·         Line 265: “These results suggest that restoring of protein prenylation is due to […] generated by the stress, which would allow to compete with the inhibition and permit membrane localization again.” In this case the ‘stress’ is a chemical treatment inducing substrate production, and would be more clearly stated as the treatment. “which would allow to” requires a restatement of the subject – “which would allow [what] to compete”. Finally, the excess production does not necessarily ‘compete’ with the inhibition, but could overcome the inhibition.

·         Line 290: “To evaluate if possibly an increase in metabolic pool might interfere with the prenylation capacity, […].” This describes an experiment in which a larger prenyl substrate pool dilutes our ability to see prenylation using a fixed amount of exogenous radio-labeled prenyl substrate. I don’t believe the authors wish to argue that a larger substrate pool reduces enzyme activity.

·         Line 351-352 is not written as a question, but ends in a question mark.

·         Lines 392-393: “Overall, our observations allude that […]”  To allude is to suggest specifically indirectly. These results suggest that […]

·         Lines 604-604: “Experiences were realized in biological triplicate” likely should be “experiments were performed in triplicate”

·         Lines 624-626: “This study provided evidence that MeJA impairs the ability of PFT, […] to specifically use a protein substrate typically dedicated to its sister enzyme PGGT-I.” This is actually the opposite of the intended meaning, that MeJA allows PFT to work on a substrate typically specific to PGGT.

Author Response

Reviewer 3

In “Methyl-jasmonate functions as a molecular switch promoting cross-talk between pathways for the biosynthesis of isoprenoid backbones used to modify proteins in plants,” Chevalier et al. observe that methyl jasmonate can overcome the inhibition of protein prenylation by fosmidomycin. Through the use of various chemical inhibitors, chemical inducers, and mutant lines, they propose a model in which jasmonic acid signaling loosens substrate specificity of the enzyme protein farnesyl transferase.

An excess of farnesyl diphosphate can also achieve similar results. These results seem to be of interest and worth publishing, but I believe require major revisions for interpretation.

We would like to thank the reviewer for his time and constructive comments. We will try to address every point.

Conceptual/interpretation issues:

  • If I interpret correctly, there are two mechanisms by which the GFP-CVIL gene can be prenylated in the absence of its preferred prenyl partner geranylgeranyl diphosphate (GGPP) or the prenylating enzyme Protein GGPP Transferase (PGT). In both cases, it seems that the protein is prenylated with farnesyl diphosphate (FPP) by Protein Farnesyl Transferase (PFT). In the first mechanism, this happens through an abundance of FPP (Figure 5E, Figure 6AB, reference 38). In the second, MeJA allows PFT activity despite a lack of increased FPP (Figure 5A, lines 331-335). This distinction is not made clearly in the text. In fact, Figure 3 and discussion of it (lines 265-270) suggest that MeJA may increase FPP levels, which is later argued against. If I interpreted correctly the authors should do some extensive rewriting to make this distinction clear.

To answer this statement, a purification and the identification of the prenyl group attached to GFP-C(VIL) would be necessary. Unfortunately, we were not able to perform this experiment with cells/plants treated with MeJA/MVA in a reproducible way.

Reviewer 2 pointed the same problem and we explained:

“In fact we are trying since many years to address this, until now, unsolved problem. The analysis and identification of prenyl modifications is very tricky for several reasons.  1. How to be sure that a farnesyl of a geranylgeranyl group was linked to a protein and not just interact with this protein. 2. Most of the proteins are quite hydrophilic, the isoprenyl group is hydrophobic. Very often, prenylated peptides are just lost and analyses are not reproducible: sometimes we get it, sometime not. 3. In addition, plants are rich in many different isoprenyls making the analysis even more difficult.

Most of the techniques used to analyze prenyl groups do not consider these points. There are techniques tagging proteins via alkyne-modified prenyl substrate feeding, but these techniques do no consider regulations like those we propose in this manuscript.

For all these reasons, this point cannot be addressed here, but we are convinced that solutions will be proposed in the near future.”

At present we cannot be sure that FPP is used to modify GFP-CVIL under restrictive conditions (either in the absence of plastidial GGPP, or in the absence of PGGT). It has to be mentioned that alendronate, an inhibitor described as blocking farnesyl diphosphate synthase is inhibiting the MeJA-induced relocalization. But it is also possible to debate the specificity of this drug.

In lines 300-302, the authors propose that, because radiolabeled FPP is incorporated to GFP-CVIL at a higher level than radio-labeled GGPP, the primary prenylation should be farnesylation via PFT. This is inconsistent with the results that fosmidomycin (GGPP inhibitor) blocks membrane localization more than mevinolin (FPP inhibitor), and with the result that pft mutants are able to correctly localize prenyl-substrate GFP while pgt mutants are not. Those results indication that GGPP should be the important incorporated prenyl, but the GGPP incorporation in figure 4B is so weak that to make conclusions based on a reduction or lack of reduction does not seem welljustified. This conflict should be fully explored, and perhaps the GGPP incorporation assay repeated with a greater level of radiolabeled sustrate or higher exposure, to justify any conclusions from either assay.

Under standard conditions, the MEP pathway is responsible of modifying GFP-CaM-CVIL in tobacco BY-2 cells. This can be observed by the delocalization of the GFP-labeling into the nucleus following fosmidomycin treatment. To determined, which enzyme catalyzes the reaction, which used Arabidopsis KO mutants where only one enzyme is active. The expression of GFP-CaM-CVIL showed that a plant lacking PGGT, where only PFT remains, the protein is localized in the nucleus (Figure 2). Treatments with MeJA (Figure 2), but also MVA (Figure 5), and Fol (Figure 4), indicate that PFT can modify the protein under certain conditions. It would be interesting to know which prenyl diphosphate substrate PFT is using under those conditions.

We agree that Figure 4b is not a prove for a use of FPP.  That’s why in line 306 we wrote:

“It is therefore not possible to conclude whether the protein is modified by FPP or GGPP.”

To avoid any confusion, we reorganized and deleted the next sentence: “….., pointing towards a possible modification with MVA-derived FPP.”

  • The authors rely on a difference between membrane localization and nuclear localization, but do not explicitly test membrane localization vs cytosolic localization. As the cytosol is generally held to cell periphery by the vacuole, these can be hard to separate by microscopy. Can the authors test that the MeJAinduced ‘membrane localization’ is truly a restoration of prenylation, and not some inhibition of nuclear localization, leading to protein accumulation in membrane-adjacent cytosol?

The prenylation sensor has been designed so that modified protein targets specifically the plasma membrane and no other type of membranes, like ER, etc (Gerber et al., 2009). The plasma membrane localization cannot only be allocated to the presence of the prenyl group, but the protein sequence upstream the CaaX motif is important. Both the calmodulin and the ROP6 contain a basic domain. These positive charged amino acids allow the interaction with the electrostatic field of the plasma membrane obtained by the enrichment of anionic phospholipids (phosphatidylinositol-4-phosphate, phosphatidic acidic and phosphatidylserine). Furthermore, in BY-2 cells, the vacuole occupies around 90% of the cell volume. The cytosolic localization of a protein can well be visualized by confocal microscopy (picture bellow).

The vacuole is not just one large compartment, but cytoplasmic trabeculae/strands running through, can be observed. In addition, in the absence of a nuclear exclusion sequence if the molecular mass is smaller as 60 kDa, the protein labels as well the nucleus as shown on this picture. However, it has to be noted that in many pictures, the nuclear membrane is labeled as well, suggesting that the protein does to some extent also interact with the ER. Interestingly, we observed a similar situation when we express GFP-CaM-CVIM instead of GFP-CaM-CVIL in tobacco cells, a protein substrate of PFT.

Data display and visualization issues:

  • I don’t have access to Supplemental Figure 1, so I cannot be sure exactly what it contains, but I would like to know the full list of compounds tested for ability to rescue FOS-induced mislocalization, and their effects. This would help to evaluate the significance of the MeJA result.

We set up a list of 120 conditions. Among hits, the most MeJA was the most-relevant and efficient and we decided to focus on this compound. Within the screen we identified and classified

However, at this point, we do not wish to list the identity of all detected hits. That’s why we focused on those, which were relevant for this study: MeJA, ethephon, MVA, Fol, cellulase.

A supplementary information file has been added and updated to show which hormones have been tested and what results have been obtained.

Figure 1: The red-green color scale in panel B will cause issues for readers with red-green colorblindness. Please replace with a color-blind friendly scale, or better yet show value and distribution as in panel B.

We have replaced the red and green colors with a color-blind friendly scale.

Was the experiment in panels C and D replicated? Are there statistical differences among these treatments?

The experiments shown in panel C and D are not replicated, but n>300 cells were counted for each condition measured on the same set of initial cells. Legend to the figure was  modified accordingly.

  • Figure 2: was this experiment replicated? Are these results consistently observed?

Accordingly, Figure S2 has been added to the supplementary material demonstrating reproducibility over years.

  • Figure 3: was this experiment replicated? Are these results consistently observed?
  • The determination of FOS100/MV5 is not clear from the text.

A supplemental figure showing the concentration combinations tested would help readers to understand the significance of pathway inhibition at these concentrations.

Accordingly, Figure S3 has been added to explain how these concentrations have been determined and to show reproducibility of our experiments.

  • Figure 4 panel A microscopy images lack scale bars. For both panels A and B was this experiment replicated? Are these results consistently observed? For panel B, as stated in conceptual issues, the GGPP bands seem too faint to justify strong conclusions. Is this an issue with exposure?

Moreover, if the goal is to observe differential prenylation of the substrate, the authors should also present a control for stable levels of GFP protein itself.

Scale bars have been added to figure 4A. Reproducibility of Figure 4A can be found in Figure S2 now available in the supplementary material and illustrating pictures taken over several years.

The blot obtained in Figure 4B as such has not been repeated because it represents a negative result. The only thing we can conclude from it is that PFT activity is higher than PGGT activity using GFP-CaM-CVIL protein substrate. We cannot explain why prenylation with GGPP is so low, as compared to the modification with FPP. It has to be noted that GFP-CaM-CVIL and GFP-CaM-CVIL are not expressed in tobacco cells. They are added to the reaction mixture as purified recombinant proteins in order to serve as enzyme substrate. Each well contain an identical amount of the same batch of recombinant protein. The pipetman’s variability could eventually be a factor, but the lab routinely examines sets.

This blot has been removed.

  • Figure 6: For panel A, the concentration of MVA used (2 mM) is more than enough to restore membrane localization with or without MeJA according to panel B. A more representative concentration would be better. For panel B, as in Figure 1, the authors should avoid a red-green color scale, and it would be better to show the data itself in the style of panel A or Figure 1. For both panels: was this experiment replicated?

Are these results consistently observed?

  • Lines 382-385 indicate that MV does not interfere with membrane localization in the absence of FOS and that this is shown in figure 6A. However, Figure 6A does not include MV or MV+MeJA treatments alone without FOS.

We did not include MV treatments alone in these sets of experiments, because, MV does not delocalize the protein (see Gerber et al., 2009, Hartmann et al., 2015, Huchelmann et al., 2016). Because, MV has no incidence, we did not check the effect of MeJA on MV-treated cells. However, we added a picture of MV-treated cells in the supplementary material taken in the same period of time, which appears now as Figure S4. As for Figure 1, we added that more than 300 cells have been evaluated.

Line 80: in cellula is not a common term, it seems that in vivo would be better in this case

Has been replaced

Line 125: HMGR has not been defined as an acronym or given context as a key MVA pathway enzyme.

Has been added.

Line 197,202, etc: Col0 should be Col-0

Have been replaced

Line 200: plantlets generally refers to plants regenerating from tissue culture or clonal propagation, while these are Arabidopsis seedlings

Have been replaced

Line 241: “path7ways” should be “pathways”

Has been corrected

Line 275: ‘Fol’ abbreviation has not yet been defined (farnesol)

Has now been defined “Yet, in this context, it has been shown that farnesol (Fol), known as an enhancer of plant 3-hydroxy-3-methyl glutaryl coenzyme A reductase (HMGR) activity, the key enzyme of the MVA pathway”

Line 312: “donner” should be “donor”

Has been replaced

Line 342: @HMGR2cat I believe should be α-HMGR2cat

@HMGR2cat has been replaced by “a polyclonal antibody raised against the catalytical domain of tobacco HMGR2”.

Line 356: ggb::CVIL – the : or :: are usually used to indicate a promoter and the gene whose expression it drives. This mutant line expressing a protein of interest should be written ggb GFP or similar

We used this way to describe our lines, because it is used by other teams. Among those we can cite

https://www.nature.com/articles/s41598-019-49590-3                 Figure 3b           

https://onlinelibrary.wiley.com/doi/full/10.1111/tpj.13099           Figure 6b

However, we removed the double punctuation :: to describe our transformed mutant plants.

Line 386,390: Synergy refers to a situation where the whole is greater than the sum of the parts. I do not see evidence for synergy necessarily as the combined effect of MeJA and MVA may be additive

Have been modified

Line 477: “VIGS-silencing” VIGS is an acronym for virus-induced gene silencing, the word “silencing” does not need to be repeated

Has been deleted

Comments on the Quality of  english Language

The paper, while readable and interesting, contains many instances of odd word usage that impede the meaning. An incomplete list of examples includes

  • Line 227: “MeJA endorses protein prenylation capacity in tobacco” To endorse is to give approval. Enhances may be a better choice

Has been modified

  • Line 265: “These results suggest that restoring of protein prenylation is due to […] generated by the stress, which would allow to compete with the inhibition and permit membrane localization again.” In this case the ‘stress’ is a chemical treatment inducing substrate production, and would be more clearly stated as the treatment. “which would allow to” requires a restatement of the subject – “which would allow [what] to compete”. Finally, the excess production does not necessarily

‘compete’ with the inhibition, but could overcome the inhibition.

Has been modified

  • Line 290: “To evaluate if possibly an increase in metabolic pool might interfere with the prenylation capacity, […].” This describes an experiment in which a larger prenyl substrate pool dilutes our ability to see prenylation using a fixed amount of exogenous radio-labeled prenyl substrate. I don’t believe the authors wish to argue that a larger substrate pool reduces enzyme activity.

The sentence has been modified

  • Line 351-352 is not written as a question, but ends in a question mark.

Has been removed

  • Lines 392-393: “Overall, our observations allude that […]” To allude is to suggest specifically indirectly. These results suggest that […]

Has been replaced

  • Lines 604-604: “Experiences were realized in biological triplicate” likely should be “experiments were performed in triplicate”

Has been replaced

  • Lines 624-626: “This study provided evidence that MeJA impairs the ability of PFT, […] to specifically use a protein substrate typically dedicated to its sister enzyme PGGT-I.” This is actually the opposite of the intended meaning, that MeJA allows PFT to work on a substrate typically specific to PGGT.

Impairs has been replaced by rearranges

Round 2

Reviewer 3 Report

Comments and Suggestions for Authors

Answers to reviewers

Reviewer 3

In “Methyl-jasmonate functions as a molecular switch promoting cross-talk between pathways for the biosynthesis of isoprenoid backbones used to modify proteins in plants,” Chevalier et al. observe that methyl jasmonate can overcome the inhibition of protein prenylation by fosmidomycin. Through the use of various chemical inhibitors, chemical inducers, and mutant lines, they propose a model in which jasmonic acid signaling loosens substrate specificity of the enzyme protein farnesyl transferase.

An excess of farnesyl diphosphate can also achieve similar results. These results seem to be of interest and worth publishing, but I believe require major revisions for interpretation.

We would like to thank the reviewer for his time and constructive comments. We will try to address every point.

Conceptual/interpretation issues:

· If I interpret correctly, there are two mechanisms by which the GFP-CVIL gene can be prenylated in the absence of its preferred prenyl partner geranylgeranyl diphosphate (GGPP) or the prenylating enzyme Protein GGPP Transferase (PGT). In both cases, it seems that the protein is prenylated with farnesyl diphosphate (FPP) by Protein Farnesyl Transferase (PFT). In the first mechanism, this happens through an abundance of FPP (Figure 5E, Figure 6AB, reference 38). In the second, MeJA allows PFT activity despite a lack of increased FPP (Figure 5A, lines 331-335). This distinction is not made clearly in the text. In fact, Figure 3 and discussion of it (lines 265-270) suggest that MeJA may increase FPP levels, which is later argued against. If I interpreted correctly the authors should do some extensive rewriting to make this distinction clear.

To answer this statement, a purification and the identification of the prenyl group attached to GFP-C(VIL) would be necessary. Unfortunately, we were not able to perform this experiment with cells/plants treated with MeJA/MVA in a reproducible way.

Reviewer 2 pointed the same problem and we explained:

“In fact we are trying since many years to address this, until now, unsolved problem. The analysis and identification of prenyl modifications is very tricky for several reasons. 1. How to be sure that a farnesyl of a geranylgeranyl group was linked to a protein and not just interact with this protein. 2. Most of the proteins are quite hydrophilic, the isoprenyl group is hydrophobic. Very often, prenylated peptides are just lost and analyses are not reproducible: sometimes we get it, sometime not. 3. In addition, plants are rich in many different isoprenyls making the analysis even more difficult.

Most of the techniques used to analyze prenyl groups do not consider these points. There are techniques tagging proteins via alkyne-modified prenyl substrate feeding, but these techniques do no consider regulations like those we propose in this manuscript.

For all these reasons, this point cannot be addressed here, but we are convinced that solutions will be proposed in the near future.”

At present we cannot be sure that FPP is used to modify GFP-CVIL under restrictive conditions (either in the absence of plastidial GGPP, or in the absence of PGGT). It has to be mentioned that alendronate, an inhibitor described as blocking farnesyl diphosphate synthase is inhibiting the MeJA-induced relocalization. But it is also possible to debate the specificity of this drug.

Thank you for this interesting information. I had indeed misinterpreted an FPP addition that is not addressed here and some updates to the text make this clearer, as well as proposed link between MeJA, MVA metabolite level, and PFT substrate specificity/activity.

In lines 300-302, the authors propose that, because radiolabeled FPP is incorporated to GFP-CVIL at a higher level than radio-labeled GGPP, the primary prenylation should be farnesylation via PFT. This is inconsistent with the results that fosmidomycin (GGPP inhibitor) blocks membrane localization more than mevinolin (FPP inhibitor), and with the result that pft mutants are able to correctly localize prenyl-substrate GFP while pgt mutants are not. Those results indication that GGPP should be the important incorporated prenyl, but the GGPP incorporation in figure 4B is so weak that to make conclusions based on a reduction or lack of reduction does not seem welljustified. This conflict should be fully explored, and perhaps the GGPP incorporation assay repeated with a greater level of radiolabeled sustrate or higher exposure, to justify any conclusions from either assay.

Under standard conditions, the MEP pathway is responsible of modifying GFP-CaM-CVIL in tobacco BY-2 cells. This can be observed by the delocalization of the GFP-labeling into the nucleus following fosmidomycin treatment. To determined, which enzyme catalyzes the reaction, which used Arabidopsis KO mutants where only one enzyme is active. The expression of GFP-CaM-CVIL showed that a plant lacking PGGT, where only PFT remains, the protein is localized in the nucleus (Figure 2). Treatments with MeJA (Figure 2), but also MVA (Figure 5), and Fol (Figure 4), indicate that PFT can modify the protein under certain conditions. It would be interesting to know which prenyl diphosphate substrate PFT is using under those conditions.

We agree that Figure 4b is not a prove for a use of FPP. That’s why in line 306 we wrote:

“It is therefore not possible to conclude whether the protein is modified by FPP or GGPP.”

To avoid any confusion, we reorganized and deleted the next sentence: “….., pointing towards a possible modification with MVA-derived FPP.”

Thank you for this update. This paper now seems consistent. It would be very interesting to see in future papers what might have caused the blot discrepancy !

· The authors rely on a difference between membrane localization and nuclear localization, but do not explicitly test membrane localization vs cytosolic localization. As the cytosol is generally held to cell periphery by the vacuole, these can be hard to separate by microscopy. Can the authors test that the MeJAinduced ‘membrane localization’ is truly a restoration of prenylation, and not some inhibition of nuclear localization, leading to protein accumulation in membrane-adjacent cytosol?

The prenylation sensor has been designed so that modified protein targets specifically the plasma membrane and no other type of membranes, like ER, etc (Gerber et al., 2009). The plasma membrane localization cannot only be allocated to the presence of the prenyl group, but the protein sequence upstream the CaaX motif is important. Both the calmodulin and the ROP6 contain a basic domain. These positive charged amino acids allow the interaction with the electrostatic field of the plasma membrane obtained by the enrichment of anionic phospholipids (phosphatidylinositol-4-phosphate, phosphatidic acidic and phosphatidylserine). Furthermore, in BY-2 cells, the vacuole occupies around 90% of the cell volume. The cytosolic localization of a protein can well be visualized by confocal microscopy (picture bellow).

The vacuole is not just one large compartment, but cytoplasmic trabeculae/strands running through, can be observed. In addition, in the absence of a nuclear exclusion sequence if the molecular mass is smaller as 60 kDa, the protein labels as well the nucleus as shown on this picture. However, it has to be noted that in many pictures, the nuclear membrane is labeled as well, suggesting that the protein does to some extent also interact with the ER. Interestingly, we observed a similar situation when we express GFP-CaM-CVIM instead of GFP-CaM-CVIL in tobacco cells, a protein substrate of PFT.

Thank you for the image. I am now convinced of PM localization.

Data display and visualization issues:

· I don’t have access to Supplemental Figure 1, so I cannot be sure exactly what it contains, but I would like to know the full list of compounds tested for ability to rescue FOS-induced mislocalization, and their effects. This would help to evaluate the significance of the MeJA result.

We set up a list of 120 conditions. Among hits, the most MeJA was the most-relevant and efficient and we decided to focus on this compound. Within the screen we identified and classified

However, at this point, we do not wish to list the identity of all detected hits. That’s why we focused on those, which were relevant for this study: MeJA, ethephon, MVA, Fol, cellulase.

A supplementary information file has been added and updated to show which hormones have been tested and what results have been obtained.

Thank you for the supplemental file. I understand not wishing to share the full screen and this list of hormones sufficiently shows the significance of MeJA to my eyes.

Thank you for addressing my comments on figures and minor/grammatical errors, particularly the addition of supplemental figures. I am satisfied with all of these updates. I have some few further minor text edits for the revised text:

Lines 128-131: A sentence has been split in two, but this resulted in a sentence fragment lines 128-129.

Line 203: repeated "." and "Col-0"

Line 207: epidermic should be epidermal

Lines 354-355: reference old red-green color scale immediately before describing new yellow-green color scale

Line 358: Should have a new line before "To support the idea that MVA is a link [...]"

Line 405: "tryptophane" should be "tryptophan"

Comments on the Quality of English Language

I appreciate the English language editing in this revised version of the manuscript; I believe the scientific meaning is now much clearer. There are still minor remaining clarity issues, some of which I highlighted in my comments above.

Author Response

Dear Reviewer,

thank you for your comments.

Recommended modifications have been done and well as some other mistakes that have been identified when reading the final version.